# On the Inclusion of Adaptive Potential in Species Distribution Models: Towards a Genomic-Informed Approach to Forest Management and Conservation

Elia Vajana [1], Michele Bozzano [2], Maurizio Marchi [1,*] and Andrea Piotti [1]

1   CNR-Institute of Biosciences and BioResources, Florence Research Area, 50019 Florence, Italy
2   European Forest Institute, 08025 Barcelona, Spain
*   Correspondence: maurizio.marchi@cnr.it

**Abstract:** Ecological modeling refers to the construction and analysis of mathematical models aimed at understanding the complexity of ecological processes and at predicting how real ecosystems might evolve. It is a quickly expanding approach boosted by impressive accelerations in the availability of computational resources and environmental databases. In the light of foreseeing the effect of climate change on forest ecosystems, the branch of ecological modeling focusing on species distribution models (SDMs) has become widely used to estimate indices of habitat suitability and to forecast future tree distributions. However, SDMs are usually informed based solely on environmental data without any reference to the genetic makeup underlying responses to the environment, the possibility of exchanging variants helping to persist in situ, or the capacity to chase suitable conditions elsewhere. Among the main evolutionary processes that may complement forecasts of range shifts are local adaptation and gene flow, i.e., the occurrence of genetic variants conferring a population the optimal fitness in its own habitat and the exchange of adaptive alleles between populations. Local adaptation and gene flow could be described by indices of genetic diversity and structure, genetic load, genomic offset, and an admixture of genetic lineages. Here, we advocate for the development of a new analytical approach integrating environmental and genomic information when projecting tree distributions across space and time. To this aim, we first provide a literature review on the use of genetics when modeling intraspecific responses to the environment, and we then discuss the potential improvements and drawbacks deriving from the inclusion of genomic data into the current SDM framework. Finally, we speculate about the potential impacts of genomic-informed predictions in the context of forest conservation and provide a synthetic framework for developing future forest management strategies.

**Keywords:** species distribution models; niche modeling; reaction norms; response functions; climate change; local adaptation; adaptive landscape; genomic offset

## 1. Framework

The genetic diversity of forests, at both the phylogenetic and intraspecific levels, is a crucial resource for European forests and their capacity to cope with the current climatic crisis. In the two decades, the conservation of forest genetic resources in Europe has been coordinated through the EUFORGEN Programme, which released the Pan-European strategy for the genetic conservation of forest trees in 2015 and established a core network of dynamic conservation units [1]. This strategy was upgraded to a more holistic approach with the release of the Forest Genetic Resources Strategy for Europe in 2021 [2]. Both strategies rely entirely on the mapping of species distributions to outline conservation targets outside of the species' ecological niches, and they are meant to monitor the conservation of forest genetic resources [3]; to feed international reports, such as the State of Europe's Forests 2020 [4]; and to inform and support policy decision making.

Estimates of the productivity of existing and future forests are often based on databases of species records that are associated with the environmental characteristics observed at the occurrence sites to build species distribution models (SDMs; Figure 1a,c,e), e.g., as carried out in the landmark European Atlas of Forest Tree Species [5,6]. Those models are then supposed to guide forest management and support the establishment of new plantations in the most suitable environments. However, to the best of our knowledge, the predominant modeling approach misrecognizes intraspecific genetic diversity among the key drivers of successful establishment in areas different from the historic distributions, which could likely lead to *(i)* under- and/or over-estimating the adaptive capacity of individual populations [7], *(ii)* aprioristically assuming that a population at the core of the species distribution/ecological niche might have better chances to survive compared to marginal populations independently from its genetic composition and the environmental change foreseen [8,9], and *(iii)* misleading the identification of sources of reproductive material for both in situ evolutionary rescue and the ex situ establishment of new forests [10,11].

As a result, neglecting the usage of genetic information in ecological models may heavily affect model forecasts of forest products and services, possibly heavily penalizing the transition towards a bioeconomy. Here, we propose an evaluation of several aspects of this issue, including the need for more continental-scale projects and research efforts, and we aim to stimulate the scientific debate on the integration of genomic information into the current SDM framework in order to improve our capacity to predict climate change effects on forest ecosystems.

## 2. Historic Attempts to Inform Species Distribution Models with Genetic Information

The first examples of species distribution models in forestry date back to the mid-eighties [12]. In that period, most of the effort was limited by the lack of reliable and spatially consistent climatic data, which are necessary to characterize habitat conditions in any species spatial record [13]. Such models referred to low-resolution 'habitat suitability models' and employed BIOCLIM software [14]. The advantage of this tool was the derivation of many climatic variables using interpolation routines. Such routines became the basis for the WorldClim database more than 20 years later [15], which is one of the currently most used climatic datasets worldwide. Since WorldClim, the 1 km spatial resolution became the standard for representing environmental information in spatial modeling, despite recent findings suggesting varying optimal resolutions depending on the species analyzed, the environmental covariates used, and the topographic characteristics of the sampled sites [16]. A broad range of statistical methods, including generalized linear and additive models, boosted regression trees, neural networks, random forests, and maximum entropy models, have since been applied to species distribution modeling [13].

To date, most work predicting the impact of climate change on species distributions has been developed around the core concept of 'realized niche' (sensu Hutchinson) [13,17]. In this framework, an '$n$-dimensional hypervolume' is defined based on specific ecological factors, including biotic interactions, to describe under which conditions, and where, a species can survive. However, the observation that plasticity is widespread and can help plants to survive in habitat conditions different from those encountered within their natural distributions challenged the application of the realized niche approach in the studies predicting climate-change-driven geographic shifts [18]. Indeed, this approach forecasts future distributions based on the climatic requirements observed across a species' natural range without any reference to the species' capacity to colonize new conditions different from those experienced in the past [19]. Moreover, a small set of predictive algorithms (e.g., MaxEnt) has traditionally been applied with poor attention to overfitting, which has led to excessively complex or average models in several cases [13,20].

Despite some attempt being made to inform SDMs with intraspecific indices of genetic variation [21,22], the evolutionary potential has been largely neglected as a key factor contributing to the observed spatial patterns in species survival. This is reflected by both the use of information from neutral markers and the quality of the species records used to

feed the models, where the core of the distributions is usually well-sampled at the expense of the margins, which might host relevant adaptive variants [23–25].

The first approach that included different genetic provenances to predict plant responses to environmental heterogeneity was theorized in North America between 2000 and 2010, and it was based on common garden experiments [26–28]. In this seminal work, response and transfer functions were pooled together to merge within- and between-population variabilities in a single predictive model. The climatic characterization of both provenances and test sites was used as a covariate in the model so that both environmental and genetic effects were integrated in order to predict the response to the climate. This type of approach was rapidly transferred to other species from outside of North America; for instance, trait-based models were created using data from common garden experiments in Europe, where several species were challenged across their native range, and the results were compared with growth traits observed in National Forest Inventory Surveys [29,30]. Despite the overall increase in the performance of trait-based models compared to niche-based models, the need for a better specification of the genetic component was advocated.

In parallel to trait-based models, transfer functions and reaction norms were developed to identify deployment zones and to inform the usage of seeds for afforestation in a changing climate [31–34]. In agreement with trait-based models, the climatic divergence between provenances and test sites was the main driver of the model, along with some geographic variables, such as latitude. Trait-based modeling was applied to non-native European tree species, such as the Douglas fir [35–37], and resulted in a genetically-based model able to outperform niche-based models based on the presence/absence of data. Finally, structural equation models and multi-trait models were proposed to link traits measured at common gardens to fitness, and they were tested with encouraging results [38].

Importantly, the abovementioned approaches still use the climate as a proxy for genetic differentiation with any explicit inclusion of genetic covariates in the models. However, a recent attempt to integrate climatic and genetic information was provided by Yu et al. [39], who used single-nucleotide polymorphisms (SNPs) to delineate seed and tree breeding zones for the lodgepole pine.

## 3. Relevance and Critical Aspects of Including Evolutionary Processes into Species Distribution Models

Evolutionary processes, spanning from past migrations to contemporary responses and selective pressure, have shaped extant genetic variations in forest tree populations. Among the outcomes of spatially divergent selection and gene flow, patterns of local adaptation, which occurs when populations have the highest relative fitness at their home sites and a lower fitness otherwise [7,40], are deemed key to understanding the fate of forest tree populations in rapidly changing environments [41]. Searching for genes involved in local adaptation, whether linked to a few loci with large effects or highly polygenic, is one of the major challenges that forest geneticists are tackling today, and it has profound consequences on the management of forest genetic resources [42].

The knowledge of specific genes and/or gene networks conferring local adaptation is crucial for projecting the adaptive potential of a population through space and time (Figure 1b,d,f). The adaptive potential of a population—i.e., the genomic and epigenomic variations underlying responses to environmental heterogeneity and change—depends upon the tightness of the genotype–phenotype connection, especially when the limits of plastic responses/acclimatation are approached [43]. When the adaptive potential is low or weakened, populations are likely to enter a spiral of demographic decline, possibly leading to local extinction, especially in the case of small, fragmented, or highly isolated populations. If forest tree species are not be able to track the shifts in ecological conditions imposed by climate change, a progressive maladaptation is expected. Genetic maladaptation can be estimated by the genomic offset, which is the expected distance between the current genetic makeup of a population and the optimal genetic composition that would be required for the population to maintain optimal fitness under future environmental conditions [44,45].

The recent literature stresses the importance of including different proxies for adaptiveness in predicting the genomic offset and, eventually, the fate of populations within their natural range and in areas where SDMs foresee suitable conditions [39,46,47]. Ideally, such an effort would require a deep genomic characterization at the whole geographic range level to match the spatial scale at which SDMs usually work. However, it is likely that spatial precision will be required to be downscaled at regional or even local scales in future projections, as evidence is accumulating that natural selection can act at a high spatial resolution in forest trees [48–50]. Thanks to the combination of life history traits allowing for large within-population diversity and often massive juvenile population sizes to be preserved, forest trees usually show great adaptive potential at the microgeographic scale despite low evolutionary rates [51]. Thus, we advocate not to neglect the genomic basis of microgeographic adaptation when discussing the results of future empirical studies combining genomic information and SDMs.

Among the main life history traits of forest trees is longevity, which often relates to offsets between the expected and observed genetic consequences of disturbances. For instance, the 'paradox of forest fragmentation genetics' was proposed because the genetic structure and levels of gene flow in fragmented tree populations rarely showed results compatible with theoretical expectations [52]: since most studies on the genetic consequences of fragmentation were based on comparing adult trees in fragmented vs. continuous populations and fragmentation is recent in most forest ecosystems, the genetic comparison of plants that were established before fragmentation often resulted in the neglect of genetic outcomes actually linked with anthropogenic disturbance. Therefore, it was proposed that the focus of research should be shifted to the progeny sired in impacted landscapes for early signals to be more easily detectable in the short term [53]. In turn, the major genomic consequences of ongoing climate change should be ascertained in young generations of trees rather than in old long-lived individuals so as to assess the effects of current rather than historic selection.

Besides such critical aspects that make forest trees so peculiar when studying adaptive patterns, current research is facing the challenge of integrating all the information that can be distilled from dense genomic data into the statistical framework of SDMs so as to refine spatial projections under future climates and to improve the effectiveness of conservation strategies [47]. To this aim, it is preliminary to characterize the intimate relationship between genes and the environment as exhaustively as possible so as to outline the forest potential for adaptation under future climates [54]. The next section discusses a series of well-established ecological methods devoted to identifying adaptive variation based on genomic and environmental information.

## 4. Ecological Methods Used to Identify Adaptive Variation

Since the 2000s, high-throughput genotyping has been revolutionizing our capacity to track genetic diversity in both model and non-model species by allowing unprecedented numbers of SNPs to be discovered both genome-wide and within targeted genomic regions [55]. Refined and novel estimates of parameters of conservation concern, such as migration rates and inbreeding depression, have become feasible together with the characterization of the molecular bases of evolutionary potential [43,56]. As a consequence, the putative genetic architecture of local adaptation started to be unveiled in forest plants for the first time, including genes and genetic pathways underlining polygenic-based responses to the climate in lodgepole pine, interior spruce, and oaks [57,58]; the major genes with phenological effects in the European aspen [59]; and the major genes that confer resistance to pathogens in lodgepole and jack pines, balsam fir, and spruce [7,60].

Among the most promising methods used to detect the genomic signature of spatially divergent selection is landscape genomics, a multidisciplinary approach that links spatial modeling to population genetics and that tests intraspecific variations for associations with focal habitat features driving selection [61]. Depending on whether genotypes or allele frequencies are tested, such a genome scan for outlier SNPs is referred to as 'gene-environment

association' (GEA) or 'environmental association analysis' (EAA), respectively, the latter being more sensitive to small-effect loci [43]. Candidate genes can be searched for either up- or down-stream of the outliers within specific genomic windows or annotated based on reference genomes [62], and they can include variants with major effects involved in epistatic pathways or contributing to quantitative traits and involving multilocus adaptation [63].

Two approaches to landscape genomics exist (Table 1): the first tests loci and focal selective pressures for associations one at a time (i.e., 'univariate models'), and the second includes entire sets of loci and selective pressures in a single analysis (i.e., 'multivariate models').

**Table 1.** A short list of uni- and multi-variate models in landscape genomics. 'Genetic structure' refers to the possibility of correcting for the confounding due genetic population structure in the association models. For an extensive review of the methods, we invite the reader to refer to Rellstab et al. [61].

| | Software | Target | Genetic Structure | Notes |
|---|---|---|---|---|
| **Univariate models** | Latent factor mixed linear models (LFMMs) [64] | Major genes but also small-effect loci if allele frequencies are used [43] | A given number of latent factors is derived at the same time that the effect of environment is estimated | Can be used with genotypes or allele frequencies; linear relationship assumed by default |
| | Samβada [65,66] | Major genes | PCs [1] [67], discriminant functions [68], and global ancestry coefficients [69–71] can be included as covariates provided that some criterion is used to identify $K$ [2] and the number of PCs to be considered | Can be used with genotypes; logistic (linear) relationship assumed by default |
| **Multivariate models** | Redundancy analysis (RDA) [72,73] | Both major genes and small effect loci | No correction allowed | Can be used with genotypes or allele frequencies; linear relationship assumed by default |
| | Partial redundancy analysis (pRDA) [72,73] | Same as RDA | Same as Samβada; can also correct for geographic structures [74,75] | Same as RDA |

[1] PCs: principal components as defined based on a principal component analysis. [2] $K$: number of genetic clusters as defined based on cross-validation procedures or a $K$-means analysis.

Both approaches allow for neutral genetic structures among a set of covariates to be included in order to decrease the rate of false discoveries (FDRs), which can be due to an alignment between the ecological gradients driving selection and the population structure arising from a range expansion [76,77]. Multivariate approaches, such as redundancy analysis (RDA; Figure 1b), were observed to be less sensitive to sampling strategies, to increase statistical power by *c.* 6%, and to decrease FDR by *c.* 12% in several demographic scenarios, including refugial expansions, isolation by distance, and the island model, as well as maintaining higher true-positive rates, even in the case of weak selection when compared to univariate models (*c.* 81% vs. 51%, respectively) [63]. RDA was also demonstrated to be more effective in detecting multilocus adaptation, possibly due to its capacity to outline groups of small-effect loci covarying with environmental variations [61].

Nonetheless, combining uni- and multi-variate modeling can increase power and decrease false detection, especially in exploratory landscape genomic studies [63], and using both genotype- and allele-frequency-based tests can enhance the spectrum of detection from

major genes to multilocus adaptation [43]. Furthermore, population-differentiation-based approaches have been used to complement ecological scans for local adaptation given their ability to detect signatures of selection associated with genetic clustering or haplotype differentiation rather than environmental variation [78,79]. These approaches have allowed for candidate genes for phenology regulation to be identified in beech [62] and for genomic regions highly differentiated between populations of European aspen, which would have otherwise been neglected, to be outlined [59].

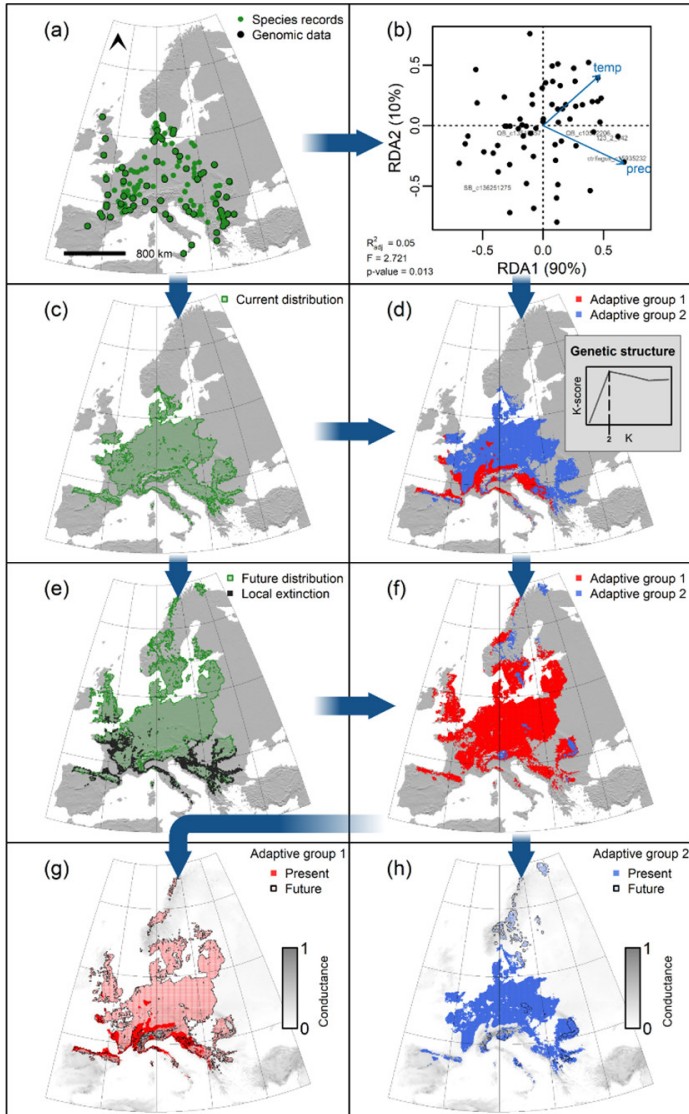

**Figure 1.** General framework integrating species distribution modeling and landscape genomics to predict the optimal spatial distribution of adaptive diversity in the future. Both approaches start from the spatial occurrences of either the species observations or the genotyped individuals or populations (**a**). A species distribution model can be derived based on the observed species records and some ecologically relevant habitat features (**c**), and it can project into the future based on specific climate change scenarios (**e**). In contrast, a landscape genomic analysis (e.g., RDA) can be devised to outline a set of candidate SNPs under selection (**b**) and to estimate the adaptive landscape of the species based on a previously inferred species distribution (**d**) [72]. The adaptive genetic structure identified can be projected into the future to outline the optimal distribution of adaptive diversity so as to tackle climate change and guarantee optimal fitness in the future (**f**). Finally, corridors for adaptive gene flow can be identified based on a conductance surface to inform forest management and conservation strategies (**g**,**h**).

## 5. Conclusions and Outlook

Characterizing genome-wide diversity will become feasible for several forest species in the next few years, at least for those taxa with a manageable genome length and complexity. While we are approaching the possibility of exploiting range-wide geographic information and finer environmental databases, advancements in analytical methods are keeping the pace in devising methods that can describe the molecular bases of adaptive potential and inform forest management and conservation under climate change on a genomic basis.

Among the most impelling challenges is conjugating range-wide predictions of genomic offsets with the shifting niches forecasted by traditional SDMs so as to understand the need for possible assisted migration strategies and gene flow (Figure 1g,h). On the one hand, methods have been proposed to account for the dispersal distances minimizing the expected genomic offset in the future but neglecting to include the potential geographical shifts in the species distributions [46]; on the other hand, geographical shifts in species distributions are usually forecasted without any reference to the genomic makeup of populations proximal to the newly suitable areas or to the potential patterns in dispersal [6]. Recent studies are indeed attempting to bridge this gap by developing comprehensive eco-evolutionary models [47,54] in the hope of providing conservation scientists with new tools to contrast the silent mass extinction of genetic diversity [80].

**Author Contributions:** All authors contributed equally to conceptualizing, writing, and reviewing this work. All authors have read and agreed to the published version of the manuscript.

**Funding:** Maurizio Marchi and Andrea Piotti have been partially supported by resources available from the Italian Ministry of University and Research (FOE-2019) under the project 'Climate Change' (CNR DTA. AD003.474).

**Data Availability Statement:** All the research data used in the present work are freely available online or obtainable directly from the authors upon request.

**Conflicts of Interest:** The authors declare no conflict of interest.

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
