# Peer review of "On the Inclusion of Adaptive Potential in Species Distribution Models: Towards a Genomic-Informed Approach to Forest Management and Conservation"

_environments, doi:10.3390/environments10010003_

Round 1

Reviewer 1 Report

Overview and general recommendation:

This article fits well with the aim and scope of the “Environments”. This review article will benefit many forestry researchers, especially those who use models.

This review article very well organized the development history of forest ecological modelling. Pro and cons of each stage are also clearly pointed out. Although the major analyzed targets focus on Europe, other areas are also considered. The whole article follows smooth logic and clear English writing. The conclusion and outlook also gave very good advice for future research. The reviewer considers the article is almost ready to publish with very tiny minor modifications.

Minor comments:

-       Line 53-60, Even though the consequences of neglecting genetic diversity in the model simulation are quite well known among the modelers, it would be nicer to offer the references after each point to support the statements here.

-       Line 126: SNP is first time mentioned here in this article, it would be better to move the “single-nucleotide polymorphism” in line 192 to line 126 unless the SNP in line 126 means something else.

Author Response

Responses to the Reviewers’ comments to

manuscript environments-1975355

submitted to Environments

COMMENTS BY REVIEWER #1

Overview and general recommendation:

This article fits well with the aim and scope of the “Environments”. This review article will benefit many forestry researchers, especially those who use models. This review article very well organized the development history of forest ecological modelling. Pro and cons of each stage are also clearly pointed out. Although the major analyzed targets focus on Europe, other areas are also considered. The whole article follows smooth logic and clear English writing. The conclusion and outlook also gave very good advice for future research. The reviewer considers the article is almost ready to publish with very tiny minor modifications.

>>> We would like to thank the Reviewer for the good appraisal of our manuscript and for acknowledging its quality and outlook.

Minor comments:

- Line 53-60, Even though the consequences of neglecting genetic diversity in the model simulation are quite well known among the modelers, it would be nicer to offer the references after each point to support the statements here.

- Line 126: SNP is first time mentioned here in this article, it would be better to move the “single-nucleotide polymorphism” in line 192 to line 126 unless the SNP in line 126 means something else.

>>> All this minor changes have been done

Reviewer 2 Report

The incorporation of genetic information to model the potential distribution of species is undoubtedly an asset to improve the quality of current information. This document proposal to be published as “Perspective” is interesting and includes a relevant bibliographic research. However, authors should explain in more detail several ideas.

In order to improve the transmission of content, I suggest that the authors present an example of how genetic information improved previously existing information. This is critical for readers to understand the scope of this document. In addition, authors should seek to validate the new information presented.

In fact, it even suggested that the authors extend the study, because the modeling of species is sometimes based on a reduced number of parameters. It should also be noted that other factors should be used more in the modeling of species, such as the intensity of solar radiation, as concluded by the following study, among others: https://doi.org/10.3390/d13080359

The concern to create more complex and rigorous models is more related to the ecological requirements of rare or endemic species, where the true ecological amplitude is sometimes poorly known. Therefore, species with large ecological ranges are of less interest from a conservation point of view.

I also do not understand why the authors end with the following sentence: “in the hope of providing conservation scientists with a new tool contrast the silent mass extinction of genetic diversity”, if articles are published daily warning of the reduction in the quality of habitats and decrease in the species distribution area. I think that “silent” is not a word adjusted to reality.

Author Response

Responses to the Reviewers’ comments to

manuscript environments-1975355

submitted to Environments

COMMENTS BY REVIEWER #2

The incorporation of genetic information to model the potential distribution of species is undoubtedly an asset to improve the quality of current information. This document proposal to be published as “Perspective” is interesting and includes a relevant bibliographic research. However, authors should explain in more detail several ideas. In order to improve the transmission of content, I suggest that the authors present an example of how genetic information improved previously existing information. This is critical for readers to understand the scope of this document. In addition, authors should seek to validate the new information presented. In fact, it even suggested that the authors extend the study, because the modeling of species is sometimes based on a reduced number of parameters. It should also be noted that other factors should be used more in the modeling of species, such as the intensity of solar radiation, as concluded by the following study, among others: https://doi.org/10.3390/d13080359

>>> Stimulated by the Reviewer’s comment, we have now included in the paper a figure (Fig. 1) and the relative text and caption (please see lines 247-258, 268-269) explaining the improvements of including genomic information into species distribution modelling, as well as a specification about the importance of data other then climatic ones which might be key for modelling species distributions (please see lines 250-251).

The concern to create more complex and rigorous models is more related to the ecological requirements of rare or endemic species, where the true ecological amplitude is sometimes poorly known. Therefore, species with large ecological ranges are of less interest from a conservation point of view.

>>> We thank the Reviewer for this suggestion, although we have to say that there are examples about widespread species which demonstrate how a proper modelling is important regardless the size of species’ distributions. The case of silver fir, which has opened a debate about its ecological niche and potential distribution between modellers and palaeobotanists (e.g. Tinner et al. 2013 doi:10.1890/12-2231.1) is a rather popular one. But also the modelling effort upon which Fig. 1 is based, showing the almost extinction in the next decades of the “blue” adaptive lineage, clearly exemplifies the potential fragility of forest genetic resources which are nowadays widespread. Please consider that the complexity of the genetic structure of forest trees itself is a key component of their potential resilience to CC. The dramatic relevance of its loss is at the basis of the message which, for instance, has been recently published in the Science journal (Exposito-Alonso et al. 2022 doi: 10.1126/science.abn5642), and it is at the core of the debate currently ongoing in Biological Conservation about the inclusion of the genetic diversity in the goals and targets of the draft Convention on Biological Diversity's (CBD) as well (e.g. Hoban et al. 2020 Biol Conserv doi: 10.1016/j.biocon.2020.108654 and the thread of literature it generated).

I also do not understand why the authors end with the following sentence: “in the hope of providing conservation scientists with a new tool contrast the silent mass extinction of genetic diversity”, if articles are published daily warning of the reduction in the quality of habitats and decrease in the species distribution area. I think that “silent” is not a word adjusted to reality.

>>> In the cited paper, recently published in Science, the Authors estimated “that more than 10% of genetic diversity may already be lost for many threatened and nonthreatened species, surpassing the United Nations’ post-2020 targets for genetic preservation” and, therefore, nicely and provocatively coined the concept of “silent extinction of genetic diversity“ to put the Anthropocene genetic diversity loss, concurrently with the habitat loss, under the spotlight.

Round 2

Reviewer 2 Report

In general, the authors read and improved all the suggested topics, which resulted in the improvement of the document. Thus, I consider that the Perspective "On the inclusion of adaptive potential in species distribution models: towards a genomic-informed approach to forest management and conservation" can continue with the process for publication in the journal Environments